# Differential Effects of Endocannabinoids on Amyloid-Beta Aggregation and Toxicity

**DOI:** 10.3390/ijms24020911

**Published:** 2023-01-04

**Authors:** Marzie Khavandi, Praveen P. N. Rao, Michael A. Beazely

**Affiliations:** School of Pharmacy, Health Sciences Campus, University of Waterloo, Waterloo, ON N2L 3G1, Canada

**Keywords:** endocannabinoids, Aβ42-induced cell toxicity, Alzheimer’s disease, neuroprotection, CB1 receptor

## Abstract

The regulation and metabolism of the endocannabinoid system has received extensive attention for their potential neuroprotective effect in neurodegenerative diseases such as Alzheimer’s disease (AD), which is characterized by amyloid β (Aβ) -induced cell toxicity, inflammation, and oxidative stress. Using in vitro techniques and two cell lines, the mouse hippocampus-derived HT22 cells and Chinese hamster ovary (CHO) cells expressing human cannabinoid receptor type 1 (CB1), we investigated the ability of endocannabinoids to inhibit Aβ aggregation and protect cells against Aβ toxicity. The present study provides evidence that endocannabinoids N-arachidonoyl ethanol amide (AEA), noladin and O-arachidonoyl ethanolamine (OAE) inhibit Aβ42 aggregation. They were able to provide protection against Aβ42 induced cytotoxicity via receptor-mediated and non-receptor-mediated mechanisms in CB1-CHO and HT22 cells, respectively. The aggregation kinetic experiments demonstrate the anti-Aβ aggregation activity of some endocannabinoids (AEA, noladin). These data demonstrate the potential role and application of endocannabinoids in AD pathology and treatment.

## 1. Introduction

The endocannabinoid system is an endogenous signaling system with complex roles, consisting of ligands known as endocannabinoids (ECs) and cannabinoid receptors such as CB1 and CB2 receptors [1,2,3]. Endocannabinoids are lipophilic molecules that are able to bind and activate cannabinoid receptors. Cannabinoid receptors are G-protein-coupled receptors (GPCRs) found throughout the body, including the central and peripheral nervous systems [4,5]. Endocannabinoids, as a class of signaling lipids, contain amides and esters with a long chain of polyunsaturated fatty acids [6]. N-Arachidonoyl ethanol amide (AEA), also known as anandamide, and 2-arachidonoyl glycerol (2-AG) are the most well-studied and best-characterized endogenous cannabinoid ligands [7]. 2-AG is the natural ligand at the CB1 receptor in the central nervous system and is found at concentrations about 100 times higher than anandamide [8]. Both anandamide and 2-AG are glycerophospholipids derived from arachidonic acid (AA), formed locally from membrane phospholipids and released from the cells in response to increases in cellular calcium [9]. Additional AA derivatives have also been identified as endogenous cannabinoids, including arachidonoyl dopamine (NADA), 2-arachidonoyl glyceryl ether (noladin or noladin ether or 2-AGE), and O-arachidonoyl ethanolamine (OAE, virodhamine) [3].

Endocannabinoids play a critical role in the central nervous system, including Alzheimer’s disease (AD), which involves the progressive degeneration of cortical and hippocampal neurons [10,11,12]. The complex pathogenesis of AD involves amyloid plaques, mitochondrial dysfunction, phosphorylation of tau protein, neurotransmitter pathway disruption, oxidative stress, and inflammation [8,10,11,12,13,14]. Among the biochemical hallmarks of AD, amyloid-β (Aβ) peptides are considered as one of the major putative pathological causes of AD [15]. Major Aβ peptides contain either 40 or 42 amino acids, and Aβ42 in particular forms toxic amyloid oligomers and plaques [13]. The pathological accumulation of Aβ begins with small numbers of monomers first aggregating into oligomers intraneuronally, which then continue to aggregate into fibrils and ultimately amyloid plaques [13,16]. Preventing oligomerization has long been a goal of preventative or therapeutic treatments for AD [8,13,14,17].

There is preliminary evidence that endocannabinoids may play a role in AD pathology by providing neuroprotection against excitotoxicity via their role in inhibiting pre-synaptic glutamate release [18,19]. Endocannabinoids act as a retrograde messenger to inhibit neurotransmitter release upon activating pre-synaptic CB1 receptors [20]. In AD, aberrant retrograde 2-AG signaling could cause synaptic dysfunction and contribute to ongoing pathology and cognitive impairment [21]. Another putative role for the endocannabinoid system is targeting inflammatory neurodegenerative processes [22]. Microglial cells and macrophages around amyloid plaques have been considered as critical elements of the inflammatory response [23]. CB2 receptors can potentially suppress microglial activation and reduce their production of pro-inflammatory molecules [24]. Additionally, it has been reported that cells lacking CB receptors are more susceptible to neurodegeneration [8,13,14,17,25]. Endocannabinoids also possess antioxidant activity and scavenge reactive oxygen species (ROS), and reduce lipid peroxidation, which could contribute to a reduction in Aβ-induced neuronal cell death [26].

This research aimed to evaluate the endocannabinoids AEA, 2-AG, NADA, noladin, and OAE for their ability to inhibit Aβ42 aggregation and to protect cells against Aβ42-induced toxicity. The anti-Aβ42 inhibition activity of endocannabinoids was investigated in vitro via thioflavin T (ThT) based fluorescence aggregation kinetics assay, and the neuroprotection of endocannabinoids was assessed in Aβ42-induced cytotoxicity in mouse hippocampal neuronal HT22 cell line. The role of CB1 receptors in endocannabinoid mediated neuroprotection was evaluated using Chinese hamster ovary (CHO) cells expressing CB1 receptors. Our results demonstrate that some endocannabinoids (e.g.,: AEA and noladin) were able to exhibit significant inhibition in Aβ42 aggregation and can also reduce Aβ42-induced cytotoxicity via CB1 receptor-mediated and non-receptor-mediated mechanisms.

## 2. Results

### 2.1. Effect of Endocannabinoids on Aβ42 Aggregation and Inhibition by ThT Fluorescence

The inhibition percentage of endocannabinoids (1, 5, 10 μM) on Aβ42 (5 μM) in aggregation kinetic assay is shown in Table 1 and Figure 1. AEA, noladin, and AA showed maximum inhibition of 93.3%, 72.9%, and 94.5%, respectively, at the highest concentration tested (10 μM) (Figure 1a–c). Other endocannabinoids, including 2-AG, NADA, and OAE exhibited much less inhibition of aggregation (Appendix A). The aggregation kinetic study of arachidonic acid (AA), which is a precursor and metabolite of endocannabinoids in vivo, demonstrates the excellent inhibition of Aβ42 aggregation (25%, 86%, 94% inhibition, respectively, at 1, 5, 10 µM) (Figure 1c).

### 2.2. Effect of Endocannabinoids Aβ42-Induced Decreases in Cell Viability

We used two cell lines to investigate the effect of the endocannabinoids against Aβ42-induced toxicity. We hypothesized that CB1 receptors might be involved in the endocannabinoid-induced neuroprotection against Aβ42 toxicity. Although HT22 cells are reported to express CB1 receptors, we were unable to detect receptor expressions via Western blot; thus, either the HT22 cells grown under our conditions do not express the receptor or do so at a level that is below detection by Western blot [27,28,29,30]. We also used CHO cells transfected with human CB1 receptors and the CB1 antagonist (AM251 at 5 µM) to further evaluate whether the effects of the endocannabinoids were CB1 receptor-dependent. All endocannabinoids were evaluated alone to see whether they exhibited any toxicity and were found to be non-toxic at the concentrations used in subsequent experiments in both cell lines. In both HT22 and CHO cells, Aβ42 (5 μM) was able to induce significant cytotoxicity (*p* < 0.0001, ****, Figure 2 and Figure 3). In HT22 cells, the protective effects of 2-AG and NADA at all concentrations tested against Aβ42-induced toxicity were small and not statistically significant. However, AEA (10 μM) was able to increase the cell viability of HT22 cells significantly (*p* < 0.05 *, Figure 2a). The CB1 antagonist (AM251) at 5 µM did not reverse the protective effects of AEA in HT22 cells (Appendix A). 

Noladin also increased the cell viability significantly (*p* < 0.05 *) in HT22 cells at 10 μM (Figure 2b), and again AM251 as a CB1 antagonist could not reverse this protection in HT22 cells (Appendix A). Additionally, OAE was able to inhibit Aβ42 aggregation significantly at (1, 10 μM) in HT22 cells (*p* < 0.05 *) (Figure 2c), and AM251 could not reverse this protection (Appendix A). AA, a common metabolite and also a precursor of endocannabinoids, showed a significant protective effect toward Aβ42-induced toxicity in HT22 cells at 5, 10 μM (*p* < 0.05, *, *p* < 0.01, **, respectively, Figure 2d). However, AM251 could not reverse this effect (Appendix A). Since we could not detect the expression of CB1 in HT22 cells, we repeated the experiments in CB1 overexpressing the CHO cell line in order to determine the role of CB1 in Aβ42-induced cytotoxicity. The protective effects of 2-AG, AEA, NADA at 1, 5, 10 µM against Aβ42-induced toxicity on the CB1-CHO cell were not significant in this cell line. 

Noladin did increase cell viability significantly (*p* < 0.05 *) in CB1-CHO cells at the highest concentration (Figure 3a), and AM251, as a CB1 antagonist, blocked this protective effect (*p* < 0.05 *, Figure 3b). OAE, at all the concentrations tested (1, 5, 10 μM), also increased the cell viability significantly in CHO cells (*p* < 0.05 *), and AM251 reversed this effect as well (*p* < 0.05 *, Figure 3c,d). Similarly, AA at all tested concentrations (1, 5, 10 µM) was able to increase the cell viability of CHO cells treated with Aβ42 (5 µM), and AM251 was able to reverse this protection (Figure 3e,f).

## 3. Discussion

This study aimed to evaluate and compare the ability of endocannabinoids to inhibit Aβ42 aggregation and toxicity using in vitro fluorescence and cell culture studies. The ThT-based fluorescence aggregation study results show that the endocannabinoids AEA and noladin exhibit excellent inhibition of Aβ42 aggregation (93% and 73% inhibition, respectively, at 10 μM), which suggests that these two endocannabinoids can directly interact and prevent Aβ42 self-assembly into higher order structures. In the previous study, these endocannabinoids also showed direct interaction with the Aβ42 peptide [31]. All endocannabinoids, as well as arachidonic acid, were tested under all protocols. In some cases, there is a consistency between amyloid aggregation and cell viability. However, in other cases, endocannabinoids like OAE are able to provide protective effects without apparent effects on amyloid aggregation, possibly via receptor-mediated effects or additional, unknown effects. In the HT22 hippocampal neuronal cells, both AEA and noladin were able to offer significant protection against Aβ42-induced cytotoxicity at 10 μM (*p* < 0.05, Figure 2), as well as OAE that was able to demonstrate neuroprotection at both 1 and 10 μM (Figure 2). In addition, the CB1 receptor antagonist, AM251, inconsistently blocked the protective effects of specific endocannabinoids.

Since oxidative stress is involved in Aβ42 toxicity, the beneficial properties of these compounds (AEA, noladin, OAE, AA) might result from reducing oxidative damage directly or activating anti-apoptotic pathways as a response [31,32]. However, other receptor-mediated and receptor-independent mechanisms should be considered [33]. AEA, noladin, and OAE have shown anti-inflammatory, antioxidant capacity, neuroprotective effects, and oxidative stress inhibition via receptor-independent pathways [34]. 

Amyloid aggregation contributes to the dysfunctional properties of neurons through other conditions, such as disrupted neurotransmission signalling and impaired endosomal-lysosomal pathways [35]. Endocannabinoids also were found to protect HT22 cells against Aβ42 neurotoxicity by upregulating PKC expression within their membranes, which indicated that PKC might play a role in the neuroprotective properties [36]. Previous evidence reported that the effects of some endocannabinoids, e.g., AEA, may involve effects not mediated by the CB1 receptor [2,34,37]. Since endocannabinoids can affect the stability of several lipid rafts, these non-CB-mediated actions could occur independently of cannabinoid receptors due to the lipophilic structure of these molecules [8,32,38]. Since AM251, a CB1 antagonist, could not reverse the effects of all endocannabinoids, our result suggests the protective effects of AEA, noladin, OAE, and AA on the HT22 could be CB1 receptor-independent. The role of endocannabinoids via non-CB receptor-mediated actions in the present study is similar to previously reported evidence and may partially involve other GPCR effects, transient receptor potential (TRP) receptors regulation, endocannabinoid metabolism via COX-2 into other biologically active compounds, as well as neuroprotective activity at PPARa receptors [39,40].

The next step in this study was looking for endocannabinoids’ impact on CHO cells expressing the human CB1 receptors, as well as using AM251, a CB1 receptor antagonist as a means to evaluate the impact of the pharmacological role of the CB1 receptor in protection against Aβ-induced toxicity. AM251 was able to inhibit the protective effects of some, but not all, endocannabinoids in CB1-CHO cells. AM251, as a CB1 antagonist, reversed the endocannabinoid protective effects of noladin, OAE, and AA, Aβ42 treated cells. The results of this study on CB1-CHO cells are consistent with several studies that reported CB1 receptor-dependent neuroprotective actions of endocannabinoids in the extraneuronal cells [41,42]. It should be noted that AA has been reported to have agonist activity for the CB1 receptor [43]. 

Additionally, CB1 and CB2 receptors activated in glia and microglia are also demonstrated to suppress elevated ERK/p38MAPK phosphorylation and COX-2 expression induced by Aβ42 [16,44]. Previously was shown that endocannabinoids such as AEA and noladin (at nanomolar concentrations) prevent Aβ peptide-induced neurotoxicity through CB1 receptors and mitogen-activated protein kinase-dependent mechanisms that can be reversed by CB1 antagonists [31]. Moreover, studies have demonstrated that the CB1Rs activation or endocannabinoid-degrading enzymes inhibition (FAAH, MAGL, and alpha/beta-hydrolase domain-containing) may enhance Aβ clearance across the blood–brain barrier by increasing the expression of the low-density lipoprotein receptor-related protein 1 (LRP1) [45]. CB1 activation in a previous transitional study showed PPARc signalling up-regulation that improves neuroinflammation, neurodegeneration, and spatial memory impairment induced by the Aβ peptide [46]. Additionally, in vivo and in vitro studies reported the neuroprotective effects of certain endocannabinoids (e.g., noladin, OAE) via CB1 receptors, may also prevent tau hyperphosphorylation [31,47].

## 4. Materials and Methods

### 4.1. Amyloid Aggregation Kinetic Assay

The endocannabinoids (AEA, 2-AG, NADA, noladin, and OAE) and arachidonic acid (AA) were obtained from Cayman Chemical Company, Ann Arbor, MI, USA and were >98% pure. The Aβ42 (>97%) was obtained from rPeptide, Bogart, GA, USA. The ThT fluorescence assay was performed in Costar, black, clear-bottom 384-well plates using Aβ42 in the absence or presence of endocannabinoids. The readings were recorded as fluorescence intensity units, obtained by measuring ThT excitation and emission at 440 nm and 490 nm, respectively. The change in excitation-emission is related to the conformational change of ThT, which is detected upon the interaction of ThT with the β-sheet formation of Aβ42 oligomers and fibrils. The data was collected every 5 min using a BioTek synergy H1 microplate reader (Agilent, Santa Clara, CA, USA) with continuous shaking at 730 cycles per minute (cpm) for 30 s, with the temperature maintained at 37 °C for 24 h. Methylene blue (MB), resveratrol, and orange G (Sigma-Aldrich, St. Louis, MO, USA, known Aβ42 aggregation inhibitors) were used as reference compounds. The assay was conducted in triplicate, and the results were expressed as percentage inhibition of Aβ42 aggregation. All compounds were prepared fresh in DMSO and diluted in Na_2_HPO_4_·7H_2_O in UPW (Cayman Chemical Company, Ann Arbor, MI, USA), adjusted to pH 7.4, with endocannabinoid concentrations of 1, 5, and 10 µM concentrations. The Aβ42 was prepared as peptide stock solutions by dissolving in 10% NH_4_OH, and further sonicated to ensure homogeneity to the final concentration of 5 µM. A 15 µM ThT stock solution was prepared using 50 mM glycine and sodium hydroxide (NaOH) buffer (adjusted to pH 7.4). The mentioned procedure and protocols adapted from Tin and coworkers [48].

### 4.2. Cell Culture & MTT Cell Viability Assay

HT22 cells (mouse hippocampal cell, a gift from Dr. Robert Cumming, PhD, University of Western, London, ON, Canada) and CB1-CHO cells (Chinese hamster ovary cells that express human CB1r, a gift from Dr. Robert Laprairie, PhD, University of Saskatchewan, Saskatoon, SK, Canada) were used in this study. The ability of endocannabinoids and arachidonic acid (>98%, Cayman Chemical Company, Ann Arbor, MI, USA) to display a reduction against Aβ42 (>97%, rPeptide company, Bogart, GA, USA) mediated neurotoxicity was determined by performing a cell viability assay. CB1 antagonist (AM251) (>98%, Cayman Chemical Company, Ann Arbor, MI, USA) was added to block CB1 receptors. Cells were cultured in DMEM and HAM’s F12 (1:1) (Thermo Fisher Scientific #SH20361, Ottawa, ON, Canada), 10% fetal bovine serum, 100 μg/mL streptomycin, and 100 U/mL penicillin. Cells were maintained in a humidified atmosphere of 95% air and 5% CO2 at a temperature of 37 °C and grew to 80% confluency (20–22 h). The complete growth media was changed every 2–5 days by trypsinizing with 0.25% trypsin/0.1% EDTA. For the MTT assay, cells were serum-starved overnight before drug treatments; then, media was exchanged for treatment media, including different compounds concentrations (1, 5, 10 µM, co-incubated with Aβ42 (5 µM) for 24 h at 37 °C). We selected 5 µM of Aβ42 and prepared oligomers based on our previous work in hippocampal neurons [17,49,50]. After endocannabinoid treatments, media was changed to serum-free, phenol red-free DMEM/F12 containing 10% MTT (thiazolyl blue tetrazolium bromide: 3-(4,5-dimethyl-2-thiazolyl)-2,5-diphenyl-2H-tetrazolium bromide). Plates were returned to the incubator for 2–4 h for the reaction to occur. Then, cells were lysed, and crystals dissolved in solubilization buffer (0.1 M HCl, 10% Triton X-100 in propane-2-ol). A Molecular Devices™ plate reader (Agilent, Santa Clara, CA, USA) was used to determine light absorption at wavelengths of 570 nm and 690 nm. The results are calculated as the percent cell viability compared to controls.

### 4.3. Western Blot

Western blots were performed as previously described to detect the CB1 receptors in CHO-CB1 cells as well as in HT22 cells [51]. Cells were scraped, sheared using 26 gauge needles, and centrifuged at 14,000× *g* for 20 min at 4 °C. Pellets were washed with phosphate-buffered saline (PBS) and lysed in chilled lysis buffer (20 mM Tris-HCl at pH 7.5, 150 mM NaCl, 1 mM EDTA, 1 mM EGTA, 30 mM sodium pyrophosphate, 1 mM β-glycerophosphate, 1 mM sodium orthovanadate, and 1% Triton X-100; and 1% Halt Protease and Phosphatase Inhibitor (Thermo Fisher, Markham, ON, Canada) prior to measuring total protein using the BCA protein assay (Thermo Fisher Scientific, Ottawa, ON, Canada). Samples were heated in 3x loading buffer (240 mM Tris-HCl at pH 6.8, 6% *w*/*v* SDS, 30% *v*/*v* glycerol, 0.02% *w*/*v* bromophenol blue, 50 mM DTT, and 5% *v*/*v* β-mercaptoethanol) for 15 min at 75 °C and 5–20 μg total protein was loaded into polyacrylamide gel wells. Proteins were separated by SDS-PAGE using electrophoresis buffer (25 mM Tris base, 190 mM glycine, 3.5 mM 44 sodium dodecyl sulfate), followed by transfer of proteins to a nitrocellulose membrane using transfer buffer (25 mM Tris base, 190 mM glycine, 20% *v*/*v* methanol). Membranes were blocked with 5% non-fat milk in Tris-buffered saline (20 mM Tris base, 150 mM NaCl, pH 7.6) plus 0.1% Tween (TBS-T) for 1 h at room temperature or overnight at 4 °C. Membranes were incubated with primary antibody (added to blocking buffer) overnight at 4 °C. Membranes were washed three times with TBS-T and then incubated with a secondary antibody (horseradish peroxidase-conjugated) in the blocking buffer for 1 h at room temperature. Membranes were washed three additional times with TBS-T. Luminata substrate was used to visualize proteins on the Invitrogen iBright 1500F imaging station (Thermo Fisher Scientific, Ottawa, ON, Canada). After imaging, membranes were probed with the primary antibody against β-actin and human CB1 receptor (Cayman Chemical Company, Ann Arbor, MI, USA). Anti-mouse (1:5000) and anti-rabbit (1:500) horseradish peroxidase (HRP) enzyme-conjugated IgG secondary antibodies were used. Western blot analysis of cell lysates of CHO cells detected the presence of a major CB1 immunoreactive bond using Fisher BioReagents™ EZ-Run™ Prestained Rec Protein Ladder, Fisher BioReagents (Catalog No. BP3603500), which was close to the expected molecular mass of the CB1 receptor (approximately 43 kDa) (Appendix A). We were unable to detect CB1 receptor expressions in HT22 cells via Western blot; thus, either the HT22 cells do not express the receptor or do so at a level that is below detection by Western blot.

### 4.4. Statistical Analysis

To assess the effects of Aβ42 (5 µM) alone or Aβ42 mixed with each endocannabinoid (1, 5, 10 µM) and Aβ42 with endocannabinoids (mentioned concentrations) mixed with AM251 (5 µM) on cell viability, a one-way ANOVA with Dunnett’s multiple comparisons test was performed to establish statistical significance between compound groups and vehicle-treated control (α = 0.05). The results are shown as the average ± standard error of the mean (SEM). The data is representative of quadruplicates samples for n = 4 independent experiments.

## 5. Conclusions

Our studies demonstrate that endocannabinoids AEA and noladin prevent Aβ42 aggregation, whereas, in the HT22 hippocampal neuronal cells, AEA, noladin, and OAE were able to rescue cells from Aβ42-induced cytotoxicity. Results from the CB1 expressing CHO cells show that the neuroprotective effects demonstrated by endocannabinoids are mediated via CB1 receptors. Further experimental studies will be necessary to directly address the question of whether endocannabinoids are able to induce neuroprotection effects via CB1 receptor and non-receptor pathways. Nevertheless, our current findings will have implications in understanding the molecule mechanisms of CB1 mediated neuroprotection in AD and in the development of novel endocannabinoid derived therapies for AD.

## Figures and Tables

**Figure 1 ijms-24-00911-f001:**
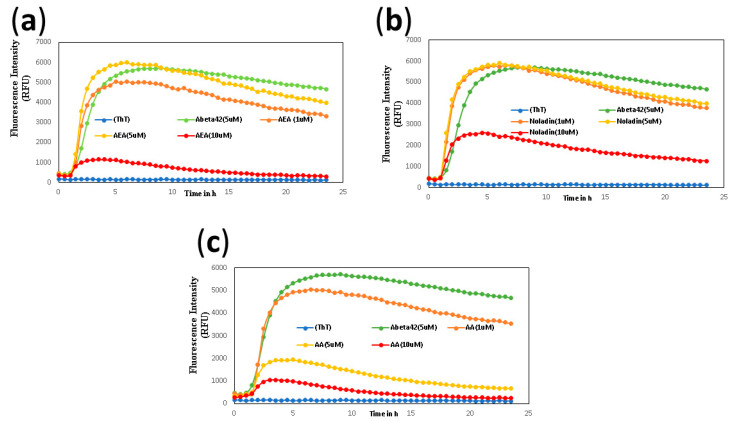
ThT-monitored 24 h aggregation kinetics of Aβ42 (5 μM) in the presence of 1, 5, 10 μM of AEA (**a**) Noladin (**b**), and AA (**c**). Aggregation kinetics were monitored by ThT-fluorescence spectroscopy (excitation = 440 nm, emission = 490 nm) at pH 7.4, 37 °C in phosphate buffer. Results are the average ± SD of three replicates.

**Figure 2 ijms-24-00911-f002:**
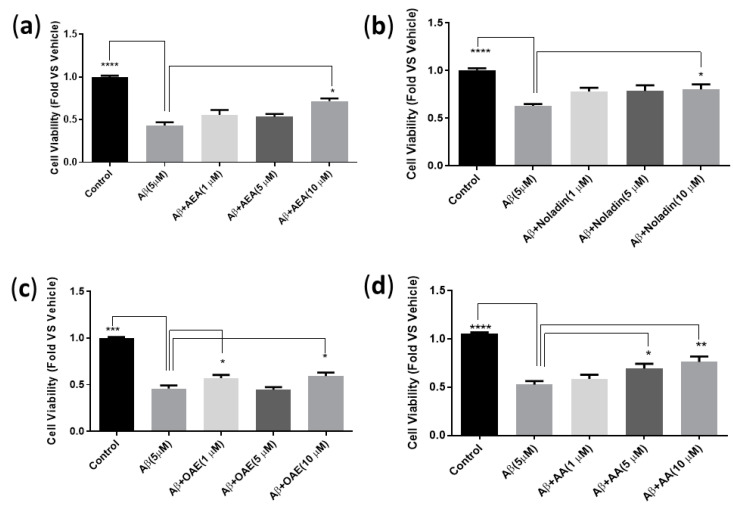
Effect of at 1, 5, 10 µM endocannabinoids and AA, against Aβ42 (5 µM) toxicity on HT22 cells. Cells were treated with Aβ42 in the absence or presence of AEA (**a**), Noladin (**b**), OAE (**c**), and AA (**d**), compared to the untreated Control group. Cell viability was determined via MTT assay. A one-way ANOVA with Dunnett’s multiple comparisons performed to establish significance between groups (α = 0.05 *, 0.01 **, 0.001 ***, 0.0001 ****). The results are shown as the average ± standard error of the mean (SEM). The data is representative 4 independent experiments, each with 4 technical replicates.

**Figure 3 ijms-24-00911-f003:**
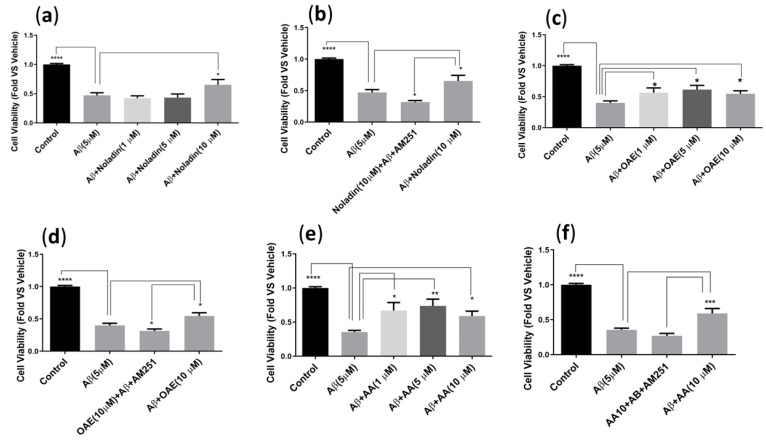
Effect of at 1, 5, 10 µM endocannabinoids and AA, against Aβ42 (5 µM) toxicity on CHO-CB1 cells. Cells were treated with Aβ42 in the absence or presence of Noladin (**a**), OEA (**c**), and AA (**e**), compared to the untreated Control group. The CB1 receptor antagonist, AM251 (5 µM), was used in panels (**b**,**d**,**f**). Cell viability was determined via MTT assay. A one-way ANOVA with Dunnett’s multiple comparisons performed to establish significance between groups (α = 0.05 *, 0.01 **, 0.001 ***, 0.0001 ****). The results are shown as the average ± standard error of the mean (SEM). The data is representative 4 independent experiments, each with 4 technical replicates.

**Table 1 ijms-24-00911-t001:** Shows the inhibition percentage of ThT-monitored 24 h aggregation kinetics of Aβ42 (5 μM) in the presence of 1, 5, 10 μM of AEA, noladin and AA at pH 7.4, 37 °C in phosphate buffer at 24 h time point. Aggregation kinetics were monitored by ThT fluorescence spectroscopy (excitation = 440 nm, emission = 490 nm). Results are average ± SD of three technical replicates.

Compounds	AEA	Noladin	AA
Inhibition Percentage (10 µM)	93.5 ± 1%	72.9 ± 15%	94.5 ± 0.6%
Inhibition Percentage (5 µM)	14.5 ± 4.3%	14.6 ± 6.8%	86 ± 5.3%
Inhibition Percentage (1 µM)	29.0 ± 8.3%	19 ± 9.2%	25.3 ± 7.5%

## Data Availability

Beazely, Mike, 2023, “Data for “Differential Effects of Endocanna-binoids on Amyloid-Beta Aggregation and Toxicity””, https://doi.org/10.5683/SP3/RC5QLF, Borealis, V1, UNF:6:r4buwsUPArwsM2qPHEA4vg== [fileUNF].

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
