# Peer review of "Differential Effects of Endocannabinoids on Amyloid-Beta Aggregation and Toxicity"

_ijms, 2023, doi:10.3390/ijms24020911_

Round 1
Reviewer 1 Report (Previous Reviewer 1)
I would like to thank the Authors and Editors for the opportunity to comment on this resubmission.
First of all, I appreciate that the Authors have changed the title of the manuscript to address the focus of their work more specifically. I have also noticed that most of the Reviewers' comments have been appropriately addressed and the text has been integrated according to their requests. I think that the improvements applied to the text (including adding the Supplementary data) are enough for this to be a publishable version of the study. However, I understand the concerns raised by the other Reviewers, specifically about the limited quantity of novel data and the lack of cell culture experiments, which may impair the overall strength of the paper. I would suggest to the Editors, therefore, that this work may be published in its current form as a Communication, instead of an Original Article.
Author Response
We thank you for your response and we have changed the format to "communication".
Reviewer 2 Report (Previous Reviewer 2)
The manuscript now entitled: “Differential effects of endocannabinoids on amyloid-beta aggregation and toxicity” by Marzie Khavandi, Praveen P. N. Rao and Michael A. Beazely has improved in overall quality and readability. I would like to thank the authors for considering the comments and try to address some of them. While I believe that the suggested additional experiments would have further improved the manuscript, the results are now supporting the title.
It is a detail, but the kinetic graphs would still be easily readable after adding the confidence intervals.

Author Response
We thank you for your response!
Reviewer 3 Report (Previous Reviewer 3)
Authors have performed minor modifications, however there are some questions unresolved:
As I mentioned before experimental part is short: There are just 2 different types of experiments (MTT assay and aggregation assay) Authors should extent experimental section and study anything else as ROS production, apoptotic markers, involvement of PKC protein, even in a primary neuronal culture.
-Authors declare that " All of the groups were serum-starved overnight before drug treatments, including the control group, which is shown as the first bar in the graphs of Figure 2,3". Considering this, authors should include in their graph one condition "control without starvation" to demonstrate that starvation is not modifying by itself cell viability.
Author Response
Re: point #1I believe the decision to alter the format of the paper from a research article to communication resolves this issue. As we have indicated previously, all of the suggestions for expanding this research are excellent, but logistically, this work was carried out by a Master’s student who has already graduated and is not available for additional experimental work.
Re: point #2
Serum starvation does indeed modify cell viability, primarily by reducing the rate of cell growth and division during the serum starvation period. However, it does so for all experimental conditions. I think a more detailed examination of the cell culture conditions would be justified if the Ht22 cells were less well-studied. However they have been used for toxicity-related assays by several groups, and our protocol is typical of labs using the cell line. We would be happy to add additional references to support the specific cell culture protocol used in this study. A few examples here: He M, Liu J, Cheng S, Xing Y, Suo WZ. Differentiation renders susceptibility to excitotoxicity in HT22 neurons. Neural Regen Res. 2013 May 15;8(14):1297-306. doi: 10.3969/j.issn.1673-5374.2013.14.006. PMID: 25206424; PMCID: PMC4107644. Liu J, Li L, Suo WZ. HT22 hippocampal neuronal cell line possesses functional cholinergic properties. Life Sci. 2009 Feb 27;84(9-10):267-71. doi: 10.1016/j.lfs.2008.12.008. Epub 2008 Dec 16. PMID: 19135458.
This manuscript is a resubmission of an earlier submission. The following is a list of the peer review reports and author responses from that submission.
Round 1
Reviewer 1 Report
I would like to thank the Authors of the Manuscript “Endocannabinoids as Amyloid beta aggregation inhibitors” for the opportunity to provide commentary on their work.
To my understanding, this study emerges from a Master’s Degree thesis and explores the ability of several endocannabinoid compounds at different concentrations on the aggregation of Amyloid beta 42 peptides and, consequently, cell viability. Two mouse cell lines have been used, one representing hippocampal cells naturally expressing the cannabinoid receptor, the other from a non-neural tissue in which CB1 receptor expression has been induced. A receptor antagonist has also been used to assess whether the putatively positive endocannabinoid effect could be non-receptor mediated. Results show that several naturally occurring endocannabinoids, as well as their fatty acid precursor arachidonic acid (AA), do have a positive effect in reducing, sometimes drastically, the aggregation of Amyloid fragments into plaques and extending cell viability, both via receptor-mediated ways and non-receptor mediated pathways (although this distinction has to be further explored experimentally).
The manuscript is very well constructed, and its contents are clear and engaging; the introduction provides sufficient background even to an unexperienced reader; the Methods section seems quite clear (I do not exactly know how to handle the fact that the used cell lines were gifted from other researchers, but I’ll take it as it is); the results are well-described and the discussion section is appropriate in framing the overall content of the work. The manuscript seems like a good starting point for further experimental analysis on the molecular mechanisms of Amyloid aggregation inhibition by ECs. My only critiques would be:
1- to provide higher quality Figures in the Manuscript, and to also provide data and figures when they are brought up in the text: declaring “Figure not shown” is highly unsettling to me, as it gives the impression that something may be purposefully hidden to the reader; if it exists, it could be put in a Supplementary file, otherwise it would be better not to bring it up in the main text.
2- To provide a full description of the Authors’ affiliation.
Reviewer 2 Report
The present work entitled “Endocannabinoids as Amyloid-Beta (Aβ) Aggregation Inhibitors” by Marzie Khavandi is, while very preliminary, interesting. However additional molecular and cellular work are required for publication in the present journal.
Comments
- Why are the standard deviation missing in the aggregation inhibition experiments? The experiment were repeated 3 time independently therefore some variability are expected.
- Line 130-131 should be in the method part
- The captions of the figures are not consistent across the manuscript.
- The many typos and repetitions give an overall bad impression of the work. e.g : in the legends: “Each column is representative of 4 replicates. The data is representative of 4 independent experiments.” Such poor wording can easily be addressed.
- While stated in the method section in the figure legend what is being display on the graph should be stated. Mean + SD preferentially.
- The tittle suggest that Endocannabinoids may inhibit AB aggregation. It is demonstrated biochemically but there is no proof from the cell culture experiments. This is necessary to support the claims.
- There is no limitation regarding supplementary information therefore rather than refereeing to data not shown the result should be provided as supplementary information. e. g.: AM251 as a CB1 antagonist could reverse this protection in HT22.
- According to the MTT results the cells might be dying. By which mechanism? Could they just stopped proliferating and be senescent? Endocannabinoids can have anti-oxidant activity that may black the AB induced oxidative stress, and following cellular response. This has to be tested.
- MTT assay relied on mitochondrial enzyme activity, could the treatment interfere with the mitochondria without being cytotoxic?

Reviewer 3 Report
The manuscript presented presented by Khavandi and colleagues study the effect of different endocannabinoids against Aβ 42 in HT22 and CHO cell lines. The manuscript includes a fairly short experimental part and some results are lost throughout the study. Some suggestions for the authors:
-authors should justify why they have selected 5 µM of Aβ 42 as toxic concentration.
-statistical analysis is missed for figure 1 and table 1. Table 1 doesn´t include replicates number.
-authors declare that “HT22 cells naturally express CB1 receptors, albeit at modest levels” (line 106). Then they say “due to the lower expression of CB1 in HT22 cells” (line 127). They should include a western-blot analysis of CB1 receptor expression in HT22 cells in their experimental conditions.
- Author should also include a western-blot to demonstrate CB1 receptor overexpression in CHO cell line in their experimental conditions.
-Why do the authors study OAE protective effect on HT22 and CHO cells if this compound has not effect on Aβ 42 aggregation. How do they explain the positive effect of 10 and 1 µM OAE on cell viability?
-there is no evidence for the statement of lines 118-119 “The CB1 antagonist (AM251) at 5 µM did not reverse the protective effects of AEA in HT22 cells”, lines 120-121 “again AM251 as a CB1 antagonist could reverse this protection in HT22”. There is no evidence about the effect of AM251 on cell viability after the treatment with AA (lines 126-127).
-authors should comment why AEA is not protective in CHO cells although this compound is able to inhibit Aβ 42 in vitro aggregation.
-experimental part is short. Authors should extent experimental section and study ROS production, apoptotic markers, involvement of PKC protein, even in a primary neuronal culture.
- in lines 249 authors declare For the MTT assay, cells were serum-starved overnight before drug treatments”. Starvation by itself could modify cell viability and results could not be related to Aβ 42 toxicity.
-there is no information about the origin of the compounds AEA, noladin… (company, reference…)
Minor comments:
-check the entire document and correct some mistakes (e.g: line 58: correct “errant” by aberrant, line 80: “inhibhitaion” by inhibition, line 86: “=” by (, line 90: delete “2.2. Figures, Tables and Schemes”
-table 1 could be organized in different way. Authors could put first the endocannabinoids that showed the best results according to inhibitory effect on ThT fluorescence assay.
-title of section 2.1 could be modified because authors have just shown the effect of cannabinoids on Aβ 42 aggregation.
-include a title before line 105 as section 2.1.
-line 108-112 should be moved to a later section in the results part. Authors should demonstrate neuroprotection of endocannabinoids first and then develop a hypothesis about the involvement of CB1 receptors.
-information of line 130-131 should be moved to material and methods section.
-lines 135-139 is the description of figure 3 and it should be remove from the results section.

Round 2
Reviewer 2 Report
Thank you for taking the time to correct the typos and fix the references.
There is no standard practice when it comes to present SD or SE. It however is much easier for the reader to see the variability of the results with SD, especially on bar graph.
The captions on the figures should be adjusted to have the same format in all the figures. Above the graph or on the left but not different location depending on the figure.
It is unfortunate that the authors did not add cell culture experiments to support the claims of the title. Fluorescence Microscopy imaging such as the one performed by Veloria et al. https://www.ncbi.nlm.nih.gov/pmc/articles/PMC6071918/ would improve the manuscript.
Alternatively, the title could be more precise and suggest that drugs reducing aggregation in biochemical tests reduce A-beta toxicity in vitro.
